# Fast 3D Surface Measurement with Wrapped Phase and Pseudorandom Image

**DOI:** 10.3390/s19194185

**Published:** 2019-09-26

**Authors:** Xing Liu, Dong He, Hao Hu, Lixin Liu

**Affiliations:** 1School of Marine Science and Technology, Northwestern Polytechnical University, Xi’an 710072, China; xingliu86@nwpu.edu.cn; 2Shenzhen ESUN Display Co., Ltd, Shenzhen 518000, China; 3Qingdao Research Institute, Northwestern Polytechnical University, Qingdao 266200, China; huhao@nwpu.edu.cn; 4School of Physics and Optoelectronic Engineering, Xidian University, Xi’an 710071, China; lxliu@xidian.edu.cn

**Keywords:** computer vision, 3D measurement, phase unwrapping, fringe projection

## Abstract

Balancing the accuracy and speed of three-dimensional (3D) surface measurement of objects is crucial in many important applications. In this paper, we present a wrapped phase and pseudorandom image method and develop an experimental system aiming to avoid the process of phase unwrapping. Our approach can reduce the length of image sequences and improve the speed of pattern projection and image acquisition and can be used as a good candidate for high-speed 3D measurement. The most critical step in our new methodology is using the wrapped phase and the epipolar constraint between one camera and a projector, which can obtain several candidate 3D points within the measurement volume (MV). The false points from the obtained candidate 3D points can be eliminated by the pseudorandom images. A systematic accuracy with MV better than 0.01 mm is achievable. 3D human body measurement results are given to confirm the fast speed of image acquisition capability.

## 1. Introduction

Fringe projection is a crucial kind of optical three-dimensional (3D) measurement technology [1] that is becoming increasingly used in various fields [2,3,4,5,6] such as industrial quality control, rapid prototyping, cultural heritage preservation, and human body 3D measurement. Unfortunately, sinusoidal fringe projection has some inevitable issues when performing such tasks relying on digital projectors, in that the measurement speed is limited by the maximum image switching rate of the projector. In order to get unambiguous phase values for increased measurement speed, the issue of phase unwrapping needs to be investigated. For this purpose, researchers try to develop different strategies for the unwrapped phase, such as gray code sequences [7], multiple spatial frequencies [8], and temporal phase unwrapping [9]. Although these methods restore the unwrapped phase with many added images recorded robustly, they still need more time to project the patterns and record images.

In order to improve the 3D measurement speed, many fast 3D surface reconstruction methods that skip the process of phase unwrapping have been proposed. Ishiyama et al. [10] were interested in using the geometric constraints between multiple cameras and projectors to overcome the obstacle of wrapped phase periodicity. The method considered the projector as an “inverse camera” and calibrated its interior and exterior parameters as added geometric constraints. Bräuer-Burchardt et al. considered the measurement volume (MV) of the system as an added constraint to obtain unique correspondence points between stereo cameras with the wrapped phase [11]. The two main steps are as follows: (1) obtain the rough 3D position with the smallest triangulation between one camera and the projector within the MV, and (2) confirm the perfect corresponding point pairs between the two cameras and rebuild the precise 3D coordinates. Guan et al. proposed a dynamic 3D imaging system with acousto-optic heterodyne fringe interferometry [12]. In this system, the fringe interferometry device is formed by the laser and acousto-optic heterodyne, and the image acquisition facility is formed by three cameras. Under the geometric constraint of three cameras, homologous points can be determined unambiguously with the wrapped phase. Kai et al. discussed a fast phase measurement profilometry system with wrapped phase, which borrowed the wrapped phase and trifocal tensor geometric constraint to obtain full spatial resolution 3D measurement, even with a minimum number of images [13].

All of the above technologies skipped the phase unwrapping step and considered the projector (or third camera) as an additional constraint. Other methods did not consider the added epipolar geometry and borrowed passive stereo matching technology to find corresponding points within wrapped phase maps. Tomislav et al. did research on combining phase-shifted and passive stereo matching, which can be divided into three main steps: (1) borrow the wrapped phase to decrease the candidate set of points, (2) use area-based matching to find the most similar match between stereo cameras, and (3) refine the initially coarse disparity map by the accurate wrapped phase value. Then, based on the refined disparity map, high-quality 3D measurements can be achieved [14]. Song et al. looked at this issue from another side, modifying fringe patterns to encode quality maps for efficient and accurate stereo matching [15,16].

The technologies mentioned above for fast 3D measurement omitting the program of phase unwrapping are classified into two categories to get rid the ambiguity of the wrapped phase: (1) calibrating the added camera (or projector), and (2) borrowing the passive stereo matching technology. The first category needs to design specific geometric arrangements of the two cameras and the projector (or added camera), which strictly restricts the valid measurement volume and sacrifices the convenience of the system. The second category tends to retrieve the absolute phase or search the matching points by using as few fringe patterns as possible. Thus, how to ensure the robustness of phase unwrapping with reduced patterns is the essential problem. In order to increase the robustness, triangulation for 3D reconstruction cannot be assembled too small, but larger triangulation will influence the correlation of passive stereo matching. Through the above analysis, our proposed methodology with a novel arrangement of two cameras and a projector is different from the typical active stereovision. There are two critical steps: (1) get candidate 3D points with the wrapped phase and the triangulation between a camera and the projector, and (2) reject the false candidate 3D points using the correlation of stereo cameras.

In order to discuss the proposed method, the rest of this paper is organized as follows: Section 2 introduces the details of the new methodology and a novel and efficient calculation method for 3D reconstruction, Section 3 presents the experimental results by the novel methodology, and Section 4 summarizes the major points drawn from this study. 

## 2. New Methodology

Before discussing the novel method, we briefly introduce conventional active stereovision, which employs a left camera, a right camera, and a digital micromirror device (DMD) projector (P) (Figure 1).

Once pairs of image coordinates from the same object point (homologous points) are identified for the already-calibrated active stereovision, the 3D surface of an object can be reconstructed by triangulation geometry. The triangulation geometry and the unwrapped phase are two significant conditions for 3D reconstruction based on the unwrapped phase, as described by Christian [17], which obtain parameters from the calibration system and fringe images recoded by cameras. Aided by the conjugate polar line, a search of the homologous points with phase values is performed along conjugate polar line segments, and the phase values only need to be considered along one direction [18]. Here, phase-shifted fringe maps are generated from the projector and cast onto the object surface. Meanwhile, the two cameras acquire the fringe images reflecting from the surface. By the phase-shifted principle, the phase obtained from the captured fringe images is wrapped and non-unique along conjugate polar line segments. The basic idea of time phase unwrapping is to vary the pitch of the fringes over time and record a sequence of phase maps. Phase unwrapping is then carried out along the time axis for one pixel independent of the others. Reducing the fringe code makes it possible to achieve shorter sequences of projection and image recording times. However, this often leads to lower robustness or accuracy. For fast 3D measurement applications, this is unacceptable. The motivation for our work was the desire to preserve measurement accuracy and robustness, with coincident reduction of the projected and recorded code length in order to realize real-time applications without extensive hardware effort.

### 2.1. Fringe Projection Profilometry of Novel Active Stereovision

Based on the typical device shown in Figure 1, in order to use the constraint of the system’s three optical components to omit the unwrapped phase, we propose a novel special arrangement for fast 3D measurement with wrapped phase.

In order to describe the proposed approach in detail, let us look at on the scheme model first, as shown in Figure 2. The novel active stereovision employs one projector, P, and a pair of cameras, C1 and C2. An arbitrary 3D point **X** in the world coordinate system is denoted by X_w_. It is written as X_c1_ in camera coordinate system 1 and Xc2 in camera coordinate system 2. When a DMD projector is working, rays departing from a point **m***_p_*(*u_p_,v_p_*) on the DMD chip plane pass through the projector lens and then irradiate point **X**, like an inverse imaging process. The irradiated point X through the camera lens is captured on the charge-coupled device (CCD) sensor plane on image point **m***_c1_*(*u_c1_,v_c1_*) in camera C1, and its corresponding image point **m***_c2_*(*u_c2_,v_c2_*) in camera C2. Moreover, the corresponding image points **m***_p_*(*u_p_,v_p_*), **m***_c1_*(*u_c1_,v_c1_*) and **m***_c2_*(*u_c2_,v_c2_*) are described as homologous image points. For the sake of simplicity, there are three optical components in the new arrangement, described as the pinhole camera model.

(1) Camera C1 and projector P create a binocular stereovision configuration for rebuilding the 3D coordinates, where $X_{w}$ and $X_{c1}$ are 3D points in the world coordinate system and camera C1 coordinate system, respectively. $R_{c1},T_{c1}$ are the rotation matrix and translation vector between the two coordinate systems, respectively.

(2) The role of camera C2 is to find point correspondences by the wrapped phase and image correlation method. The distance between cameras C1 and C2 is small, providing strong correlation between the two cameras.

First, we used the bundle adjustment strategy (BAS) [19] to calibrate the forming of stereovision by projector P and camera C1, and we can model the fringe projection profilometry (FPP) with the principle of active stereovision.
(1)$$\left\{\begin{array}{l} X_{c1}=R_{c1}X_{w}+T_{c1}\\ s_{c1}{\tilde{m}}_{c1}^{\prime }=K_{c1}{\tilde{X}}_{c1}\\ m_{c1}=m_{c1}^{\prime }+\delta \left(m_{c1};k_{c1}\right)\\ s_{p}{\tilde{m}}_{p}^{\prime }=K_{p}[R_{s}|T_{s}]{\tilde{X}}_{c}\\ m_{p}=m_{p}^{\prime }+\delta \left(m_{p};k_{p}\right) \end{array}\right.$$
where $\tilde{•}$ denotes the homogeneous coordinates, $s_{c1}$ and $s_{p}$ are the scale factors in projecting a 3D scene onto a 2D plane by homogeneous coordinates, $m_{p}^{\prime }$ is the ideal no-distortion project point, $K_{p}$ is the intrinsic parameter matrix, $\left[R_{c1}|T_{c1}][R_{p}|T_{p}\right]$ are the rotation matrix and translation vector of camera C1 and projector P, and $\delta \left(m_{•};k_{•}\right)$ is the lens distortion function of image position m and distortion coefficient $k_{p}$.

In FPP, camera C1 and projector P are fixed, and the structure parameters $R_{s}$ and $T_{s}$ of FPP are introduced to represent the rigid transformation between camera and projector, as follows:(2)$$\left\{\begin{array}{l} R_{s}=R_{p}R_{c1}^{-1}\\ T_{s}=T_{p}-R_{p}R_{c1}^{-1}T_{c1} \end{array}\right.$$

Meanwhile, the intrinsic and extrinsic parameters of camera C2 are as follows:(3)$$\left\{\begin{array}{l} s_{c2}{\tilde{m}}_{c2}^{\prime }=K_{c2}[R_{12}|T_{12}]{\tilde{X}}_{c1}\\ m_{c2}=m_{c2}^{\prime }+\delta \left(m_{c2};k_{c2}\right) \end{array}\right.$$

### 2.2 Steps to Obtain Candidate 3D Points

The above system parameters are borrowed to determine homologous points between camera C1 and projector P and rebuild the 3D points $X_{w}$ The novel method can be separated into two steps.

**Step 1.** Estimate several candidate 3D points within the MV.

The wrapped phase of camera C1 acquires four-step phase-shifted (FSPS) patterns $I_{i},\;\;i=1,\;2,\;3,\;4$, modulated by the object surface. Thus, the wrapped phase of certain valid pixels can be calculated as:(4)$$\varphi_{c1}\left(i,j\right)=\arctan\frac{I_{4}-I_{2}}{I_{1}-I_{3}}$$

In order to get the unambiguous phase value, the unwrapping phase needs to be performed and the unwrapped phase value is:(5)$$\phi_{c1}=\varphi_{c1}+2k\pi$$
where k is the unknown phase wrapped order. Figure 2 shows that the unwrapped phases of the homologous image points m_c1_ ($\phi_{c1}$) and m_p_ ($\phi_{p}$) are equal: $\phi_{p}=\phi_{c1}$. Here, we supposed that vertical fringe patterns are projected by the projector. Thus, the image coordinate $u_{p}\;$is proportional to the unwrapped phase $\phi_{p}$, and the linear relationship between the image coordinate and the unwrapped phase can be described as follows:(6)$$u_{p}=\frac{\varphi }{2\pi }•\Lambda =\frac{\phi +2k\pi }{2\pi }•\Lambda =\;\frac{\phi }{2\pi }•\Lambda +k•\Lambda \;\;\;k=0,1\ldots \left(N/\Lambda -1\right)$$
where Λ is the spatial periodic of fringes in units of pixels, N is lateral resolution, and k is the order of the wrapped phase.

By the MV constraints [17], for pixel $m_{c1}$ in camera C1, its homologous points on the projector chip are located in the region between **m***_1_*(*u_1_,v_1_*) and **m***_2_*(*u_2_,v_2_*) as shown in Figure 3. In this region, the homologous points **m***_p_*(*u_p_,v_p_*) satisfy the condition $u_{1}\leq u_{p}\leq u_{2}$. Based on Equation (5), the service candidates of $m_{ui}$ are estimated as follows: (7)$$u_{p}=\;\frac{\phi }{2\pi }•\Lambda +k•\Lambda ,\;k=n_{1},n_{1}+1\ldots n_{2}$$
where n_1_ and n_2_ are the range of order of the wrapped phase, and
(8)$$n_{1}=INT\left(\frac{u_{1}}{\Lambda }-\frac{\phi }{2\pi }\right)\;n_{2}=INT\left(\frac{u_{2}}{\Lambda }-\frac{\phi }{2\pi }\right)+1$$
where $INT\left(•\right)$ means fractions are rounded down, and the equation gives several candidate homologous points $m_{p}$.

**Step2.** Eliminate false points from the candidate 3D points.

Utilizing the epipolar constraint [17] to confirm the homologous point pairs between projector P and camera C, the corresponding wrapped phase determines the correlation peak of the corresponding wrapped phase in the region between_1_ and n_2_. Points $P_{c2,i}$ can be obtained by inserting each candidate 3D point $X_{w,i}$ into Equation (3). Different possible corresponding point pairs $P_{c1}↔P_{c2,i}\;$ exist between the two cameras. A pseudorandom image is a random intensity pattern produced by the mutual interference of a set of wave fronts. With the help of the correlation degree between camera C_1_ and C_2_ corresponding regions, false points are eliminated from possible corresponding point pairs. The correlation objective is to find the real corresponding point in camera C2 for point *P_c1_* from the possible corresponding points *P_c2,i._* The first step in this process is to shift the same size region onto the image captured by camera C2 until the correlation weight achieves the maximum, as shown in Figure 4. In the image to the right of Figure 4, the correlation region size c1 or c2 is (2w + 1) × (2w + 1) pixels.

To evaluate the correlation degree between corresponding regions in cameras C1 and C2, a correlation criterion has to be established beforehand. The normalized correlation coefficient (NCC) of each point pair is defined as
(9)$$Ncc_{i}=\frac{\sum_{c1}\sum_{c2}{\left\{\left(P_{c1}-\bar{P_{c1}}\right)*\left(P_{c2,i}-\bar{P_{c2,i}}\right)\right\}}^{2}}{{\left\{\sum_{c1}\sum_{c1}\left(P_{c1}-\bar{P_{c1}}\right)\right\}}^{2}*{\left\{\sum_{c2}\sum_{c2}\left(P_{c2,i}-\bar{P_{c2,i}}\right)\right\}}^{2}}$$
where *c1* and *c2* are the size of the correlation region of cameras C1 and C2 in the operation, and $P_{c1}^{}$$\bar{P_{c2}^{i}}$ is the mean value of the correlation region. The optimal correlation values of all possible corresponding point pairs will be selected, which correspond to real 3D points. The correlation of corresponding image regions $I_{p1}$ and $I_{p2}\;$ is remarkably strong, since the angle between cameras C1 and C2 is smaller and the perspective is similar, which drastically reduces the chance of false positives or false negatives.

### 2.3. 3D Reconstruction Algorithm

Compared with the typical method, the novel algorithm separates the process into two crucial steps: calculating the candidate 3D points and eliminating the false 3D points. This entails more computational cost. In order to improve the speed of 3D reconstruction, we present a parallel and highly efficient computing algorithm.

In a stereovision system, the projector is seen as a pinhole camera, but it is different from an actual camera. On its DMD chips, the projected phase is irrelevant to the measured object. Once the stereovision system has been calibrated by Equation (5), describing the relationship between the position of the projector retina plane and the unwrapped phase value for any pixel of the camera, its homologous point in the projector is only relevant to its unwrapped phase value. Further, one can conclude that there is a functional relationship between the phase value of the camera and the 3D coordinates. According to the Weierstrass approximation theorem, a continuous function on a closed interval can be uniformly approximated by a polynomial function as closely as possible, and the polynomial function between the 3D points $X_{w}\left(x_{w},y_{w},z_{w}\right)\;$ in the world coordinate system and the unwrapped phase $\varphi_{c}$ can be calculated as
(10)$$x_{w}\left(\varphi_{c}\right)=a_{0}+a_{1}\varphi_{c}+a_{2}\varphi_{c}^{2}\ldots +a_{n}\varphi_{c}^{n}\;y_{w}\left(\varphi_{c}\right)=b_{0}+b_{1}\varphi_{c}+b_{2}\varphi_{c}^{2}\ldots +b_{n}\varphi_{c}^{n}\;z_{w}\left(\varphi_{c}\right)=c_{0}+c_{1}\varphi_{c}+c_{2}\varphi_{c}^{2}\ldots +c_{n}\varphi_{c}^{n}$$
where $a_{0},a_{1},a_{2}\ldots ;\;b_{0},b_{1},b_{2}\ldots ;\;c_{0},c_{1},c_{2}\ldots$, the coefficients of the polynomials, are seen as a lookup table (LUT) from phase $\;\varphi_{c}\;$ to 3D coordinates $X_{w}$.

Now, we discuss how to construct the LUT. For a calibrated stereovision system made up of camera C1 and projector P, as shown in Figure 2, intrinsic parameters (principal points, optics center, lens distortion) of camera C1 are obtained, and we suppose a ray line $m_{c}^{i,j}$, which is the function of these parameters, departing from given pixel $P\left(i,j\right)$ on the imaging plane, passing through the lens. Along the ray, N sampling points $X_{w1},X_{w2},X_{wi}\ldots \;$are selected within the MV range. In order to confirm the phase value, we project the 3D point $X_{wi}\;$to the projector DMD retina plane and locate its pixel position $P_{i}\left(m_{u},m_{v}\right)$. By Equation (5), the phase of position $P_{i}$ is confirmed as $\varphi_{i}$, and each sampling point corresponds to a different phase value, as $\left(X_{w1},X_{w2},X_{wi}\ldots ↔\varphi_{1},\varphi_{2},\varphi_{i}\ldots \right)$, and some functions about the LUT, as follows:
(11)$$x_{wi}\left(\varphi_{ci}\right)=a_{0}+a_{1}\varphi_{ci}+a_{2}\varphi_{ci}^{2}\ldots +a_{n}\varphi_{ci}^{n}\;y_{wi}\left(\varphi_{ci}\right)=b_{0}+b_{1}\varphi_{ci}+b_{2}\varphi_{ci}^{2}\ldots +b_{n}\varphi_{ci}^{n}\;z_{wi}\left(\varphi_{ci}\right)=c_{0}+c_{1}\varphi_{ci}+c_{2}\varphi_{ci}^{2}\ldots +c_{n}\varphi_{ci}^{n}i=1,2,3\ldots N$$
where the sampling 3D points $X_{wi}\left(x_{wi},y_{wi},z_{wi}\right)$ and phase value $\varphi_{i}$ are known, and we make the sample number N greater than the order of polynomials n + 1, based on the least squares method. The coefficients of the polynomials $a_{0},a_{1},a_{2}\ldots$, $\;b_{0},\;b_{1},\;b_{2}\ldots$, and $c_{0},c_{1},c_{2}\ldots$ are confirmed, completing the LUT.

Related key issues regarding the new 3D reconstruction method are discussed above. Then, in order to make the whole novel process ever clearer, the details can be described as follows:
Calibrate the system. Arrange the special 3D sense device as shown in Figure 2. The active stereovision is formed by camera C1 and projector P, and camera C2 is added to determine the real 3D point from the candidate 3D points. After calibrating the active stereovision and camera C2 parameters using the BA strategy [14], using Equation (8), build the coefficients of the polynomials from phase value to 3D coordinates, described as $a_{0},a_{1},a_{2}\ldots$, $b_{0},b_{1},b_{2}\ldots$, $\;c_{0},c_{1},c_{2}\ldots$.Determine the projection and acquisition image sequence. The four-step phase-shifted (FSPS) and pseudorandom patterns are sent by the projector to the surface of the measured object, then camera C1 acquires the phase-shifted images reflected off the object. Cameras C1 and C2 simultaneously capture the pseudorandom image. The five projected image patterns are generated by computer, as shown in Figure 5.Estimate several candidate 3D points. By the phase-shifted technology in Equation (4), use the FSPS image to get the wrapped phase map of camera C1. For camera C1, any valid pixel $P_{c1}\left(i_{k},j_{k}\right)\;$ based on the relationship between phase value and pixel position (Equations (5) and (6)) can obtain every possibility of unwrapped phase $\varphi_{i}=\phi +2k_{i}\pi$ and decompose it into the LUT as Equation (8) to reconstruct several candidate 3D points.Select the true 3D points. Project all candidate 3D points to the camera C2 imaging plane and obtain candidate corresponding point pairs $P_{c1}\left(i,j\right)↔P_{c2}^{i}\left(i,j\right)$. For each point pair, check the correlation between the corresponding regions by the NCC method (Equation (7)) and select the maximum NCC value that has significant correlation. Then, its 3D point is the true one.

## 3. Experiments and Results

In order to check the measurement precision and prove the suitability of fast 3D surface reconstruction, we performed two measurement experiments.

### 3.1. Precision Test

We set up the 3D reconstruction device, as shown in Figure 6, and calibrated the system BA strategy. With the system calibration, we obtained an MV of about 300×250×80 mm. To check measuring accuracy within the MV, two different sizes of standard ceramic spheres with known radius were kept in 12 places within the MV separately. The result of the first one is shown in Figure 7. The points of spherical crowns (white) are reconstructed in different poses within the MV, and for each pose, the sphere (red) is fitted by the point cloud data, and the other spheres have similar results. An analysis of the error of the fitting sphere is shown in Table 1. In the table, the significant values of the measurement errors are explained, including maximum, minimum, mean, and standard deviation, which show that the novel method can achieve preferable measurement precision. 

### 3.2. Using the Method for Human Body Measurement

In order to state that the proposed method can quickly rebuild 3D object surfaces, we used the method for human body measurement. As shown in Figure 2, the novel arrangement consists of a digital projector (Acer H5380BD) and two GigE cameras (Imaging Source DMK 23GM021). It is important to note that in the proposed technique, the projector and one camera (C1) form the active stereovision and another camera (C2), as an assistant, helps to determine the 3D point coordinates. Then the active stereovision and assistant camera are calibrated using the BA strategy. In the experiments, cameras C1 and C2 were synchronized with the projector to acquire the FSPS method and pseudorandom images. Utilizing the discussed technology to rebuild the body surface, the results in Figure 8 show a sketch of the system. Figure 8b shows the camera C1 wrapped phase obtained by the FSPS method and the added pseudorandom images recovered by cameras C1 and C2. The result after removing the erratic points and background is shown in Figure 8c. The body 3D surface experiment explains that the proposed technology can measure complex surface characteristics of objects and is suitable for body surface measurement. 

Since there are limitations of the view and measurement volume, in order to measure the whole-body 3D surface, we describe the novel arrangement of stereovision as a depth sensor and use eight depth sensors to set up an optical measurement network. The network is calibrated by the technology in our previous work [2]. It reconstructs the measured human body from eight different views. The results are shown in Figure 9. Figure 9a shows a schematic diagram of the measurement network and Figure 9b shows different views of the body surface. The 3D point clouds of hair and shoes are missing because their colors are too dark for the 3D model to image.

In this research, the resolution of the cameras was 1280×960 and the full speed was 30 Hz in external trigger mode. We used the field-programmable gate array (FPGA) chip to control the maps projected from the projector and images acquired by the cameras. The cameras were synchronized with the projector, which refreshed the patterns at 30 Hz. Because the proposed algorithm only requires five images for 3D shape reconstruction, it only needs 1/6 s to acquire a given viewing perspective of the 3D shape of the human body. The optical measurement network can be divided into four subgroups: 1A and 2B, 1B and 2A, 3A and 4B, and 3B and 4A, Figure 9a. Each subgroup consists of two depth sensors. In each subgroup, the two depth sensors are set up in a diagonal direction and do not interfere with each other. Each subgroup projects and captures the pattern sequences simultaneously in 1/6 s. The measurement network controls four subgroups in turn to complete the whole-body 3D shape acquisition from different viewing perspectives, so it only needs about 2/3 s to finish the image acquisition. For people standing still during the less than one second image acquisition time, unconscious body shaking can be omitted. The result in Figure 9b proves that the new technology provides fast 3D shape reconstruction method for whole-body measurement.

## 4. Summary

This paper presents a novel fast 3D reconstruction technology with wrapped phase. Different from the typical active stereovision arrangement, the discussed algorithm utilizes the triangulation formed by projector P and camera C1 to reconstruct several candidate 3D points for each pixel in camera C1 with recorded wrapped phase, and further uses the correlation between camera C1 and another camera (C2) to judge the true 3D coordinated points. The main advantage of the approach is that the process of phase wrapping is omitted, so it can reduce the pattern sequence acquisition time and is suitable for application of fast 3D surface reconstruction. 

## Figures and Tables

**Figure 1 sensors-19-04185-f001:**
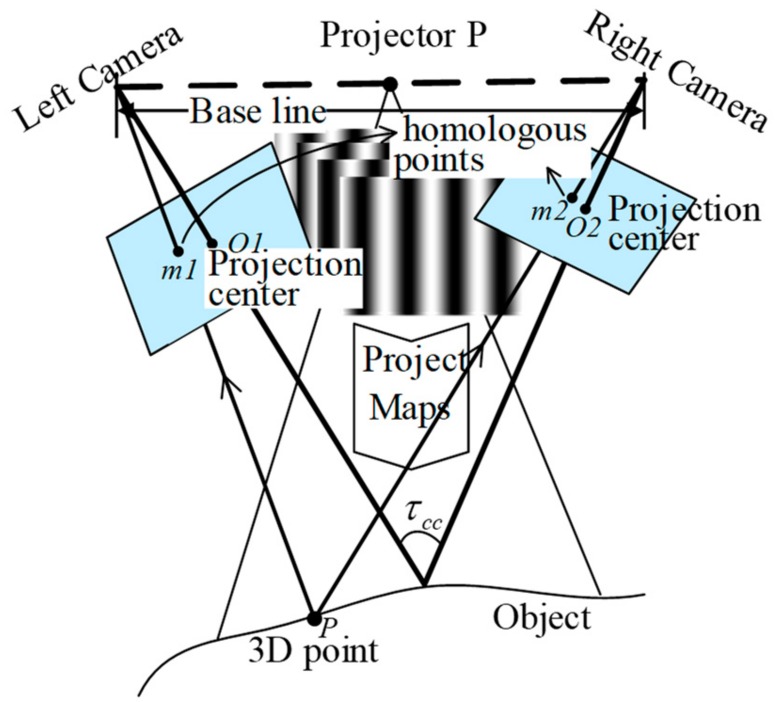
Conventional arrangement.

**Figure 2 sensors-19-04185-f002:**
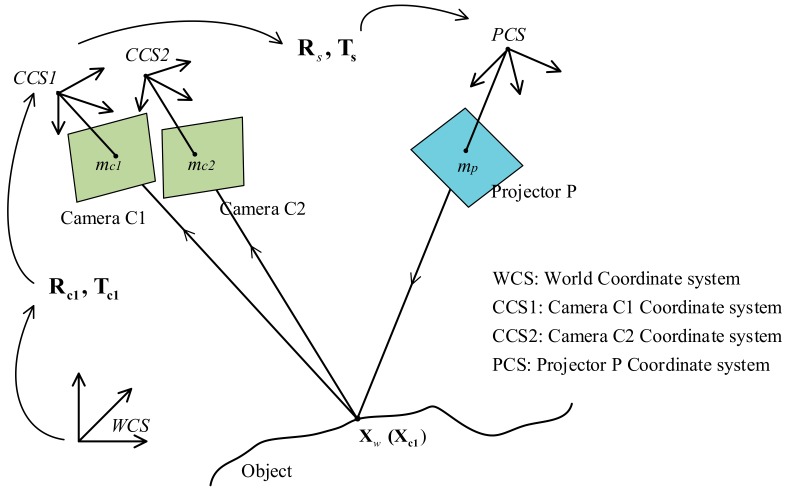
Schematic diagram of novel active stereovision scheme.

**Figure 3 sensors-19-04185-f003:**
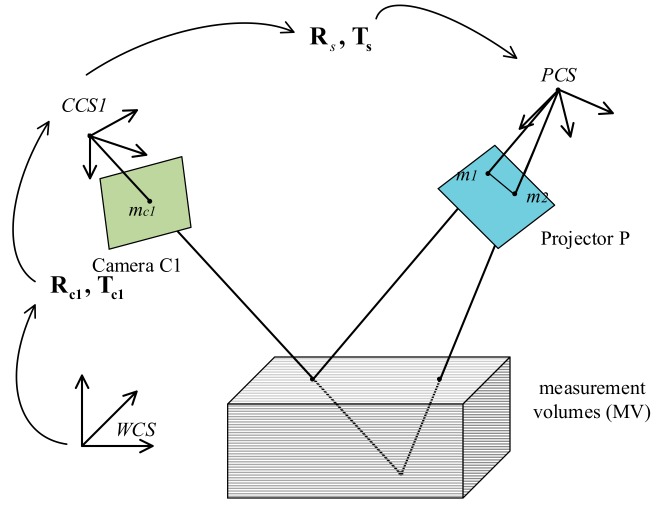
MV constraints.

**Figure 4 sensors-19-04185-f004:**
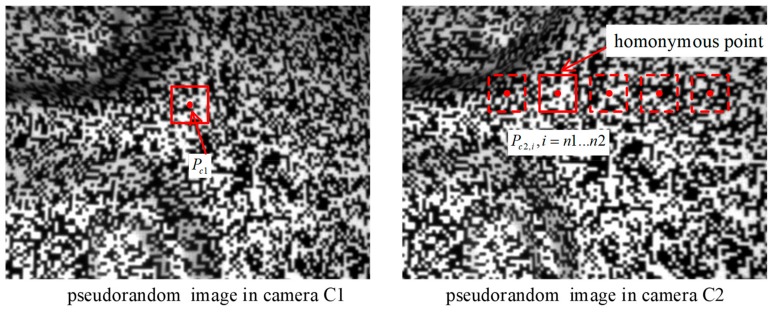
Possible corresponding point pairs between cameras C1 and C2.

**Figure 5 sensors-19-04185-f005:**
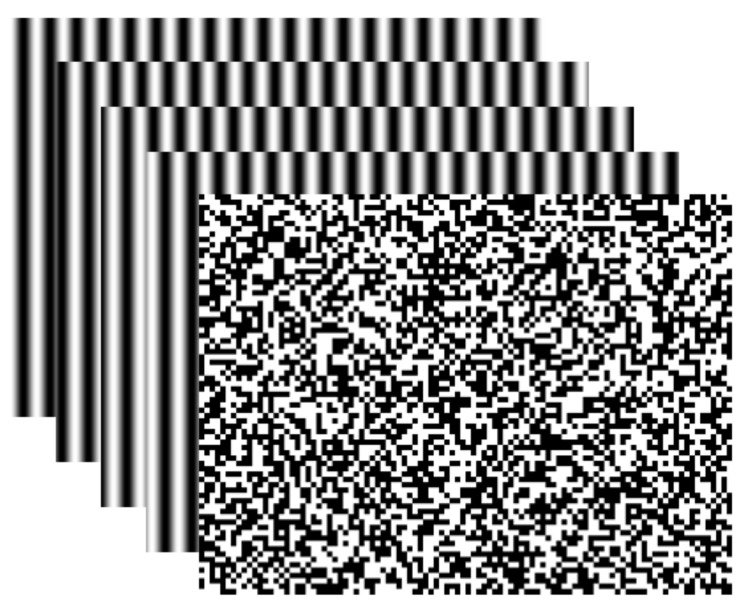
Projected image patterns.

**Figure 6 sensors-19-04185-f006:**
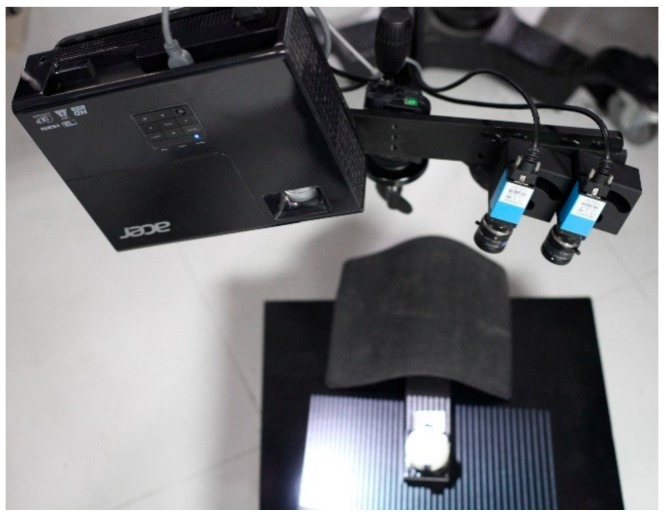
3D measurement device.

**Figure 7 sensors-19-04185-f007:**
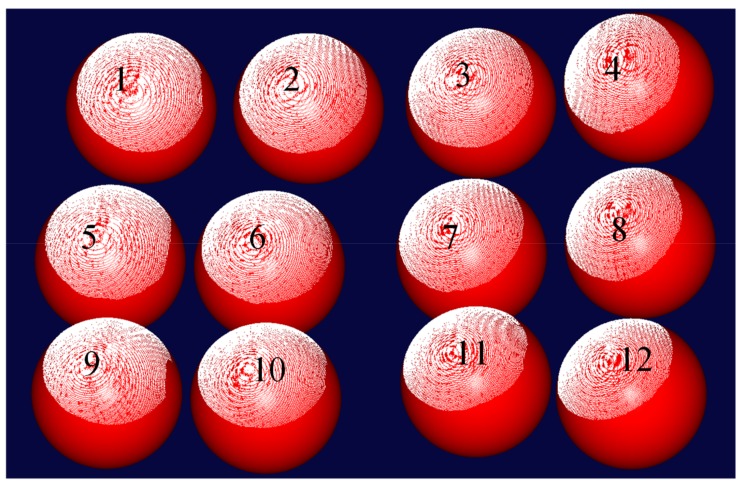
Reconstructing the spherical crown and fitting spheres in 12 different places.

**Figure 8 sensors-19-04185-f008:**
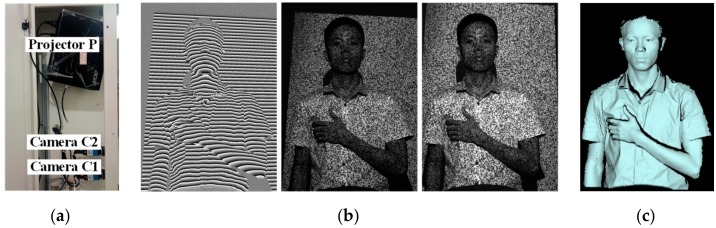
**Figure****8.** (**a**) Sketch of the system; (**b**) wrapped phase and pseudorandom images; (**c**) rebuilt 3D result.

**Figure 9 sensors-19-04185-f009:**
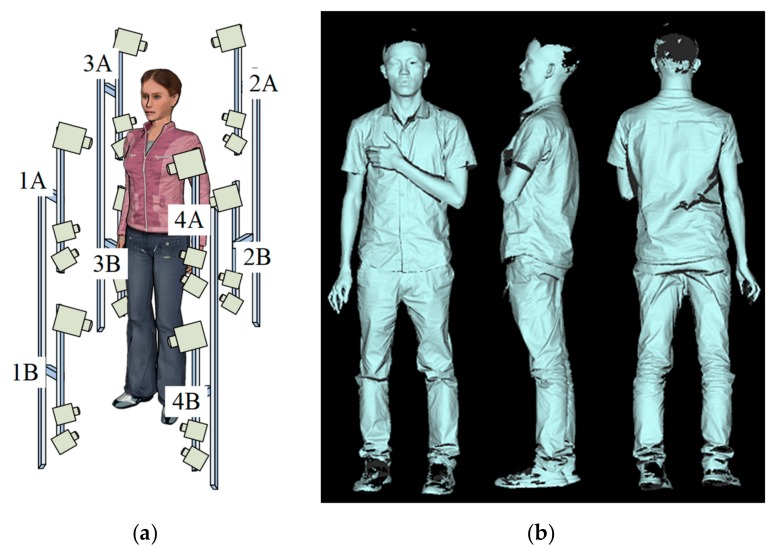
(**a**) Sketch of the measurement network; (**b**) reconstruction result of human body.

**Table 1 sensors-19-04185-t001:** Results of precision experiment. Std, standard deviation.

Nominal Value of Radius (mm)		Results of Experimental Error			
		Mean	Std.	Max.	Min.
1	25.3897	0.0103	0.0053	0.0174	0.0007
2	25.3921	0.0048	0.0045	0.0133	−0.0012

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
