# Peer review of "Fast 3D Surface Measurement with Wrapped Phase and Pseudorandom Image"

_sensors, 2019, doi:10.3390/s19194185_

Round 1
Reviewer 1 Report
Approach utilizes the triangulation formed by the projector and one camera reconstructing several candidate 3D points for each pixel in the first camera with recorded wrapped phase, and further it uses the correlation between first camera and the other one to judge the true 3D coordinated points.
The authors can reduce the length of images sequence and improve the speed of patterns projection and images acquisition, which can be used as a good candidate for high-speed 3D measurement.
There are lot errors, mainly; the authors do not use commas in the different sentences. Please revise this using help of native speaker.
Reviewer 2 Report
The text needs comprehensive checking, I wrote some suggestions up to a point (L68) but too many parts need a complete rewrite. I would suggest the authors to rewrite section 2, describing the procedure more linearly and indicating the meaning of each symbol and subscript. The differences between the proposed technique and the existing ones should be described in more detail. Captions should describe the figures in more detail. L33 "method" -> methods L36 "which skipped the process of phase unwrapping. had been proposed." -> "which skip the process of phase unwrapping had been proposed." (I removed also a point after "unwrapping"). L38 "periodic"? L40 "volume(MV)" -> "volume (MV)" L42 "obtaining the rough 3D position with the smaller triangulation": position of the camera with respect to the projector? *smallest* triangulation. L44 "precision" -> "precise"; "coordinate" -> "coordinates" L61 "looked at this issue in another side, who modify" -> "looked at this issue from another side, which modifies" L63 "Above-mentioned technology for fast 3D measurement omitted the program of phase unwrapping are classified into two categories to get rid the ambiguity of the wrapped phase: calibration the added camera (or the projector); borrowing the passive stereo matching technology": does this mean "Above-mentioned technologies for fast 3D measurement without the need of phase unwrapping are classified into two categories according to the method used to get rid the ambiguity of the wrapped phase: (1) calibration of the added camera (or the projector); (2) borrowing the passive stereo matching technology"? L68 "The second one, there are contradiction" -> "For the second one, there are contradictions" I am not sure what this sentence means "The second one, there are contradiction between the correlation of passive stereo and the robustness to noise of phase map.": can you add more detail? L100 "it spends more time to project and capture the images sequence": it needs more time with respect to what? L120 Caption of figure 2 must contain all the information to be able to understand the figure: what do the symbols and the subscripts mean? L126 eq.1: what do symbols "s_c1" and "s_p" indicate? L128 "separately"? L129-130 please explain the meaning of "homogeneous coordinate" and "intrinsic parameter matrix" L135 "are the rotation matrix and the translation vector of the camera C1 and projector P" with respect to? L139 and L152 eq. 3 and 4: please describe the parameters involved. L145 I would be useful to insert a description of how the pseudorandom images are generated. L156 I am not sure of what "the horizontal axis position m_u" is in this context. Also, what does the "u" subscript refer to? L160 please indicate what the "dot" (sorry, using a text editor without special symbols) operation is. L162 What are "mu 1 and mu 2"? L165 I do not understand: if m_ui is between m^1_iu and m^2_iu it follows that in m^k_iu the admissible values of k are only 1 or 2. Otherwise k must be a real number, not an integer one. L166 "Where "phi" is wrapped phase value in the pixel P c1 (i 1 , j 1 )" L169 - 171 does this mean that you use the epipolar to select the correct homologous point among the candidates and then eq. 1 to find the corresponding point on the surface? L180 Again, all the symbols in eq. 7 must be described, otherwise the expression is meaningless. L195 I cannot understand the sentence "Once the stereo-vision system has been calibrated, by the Eq.(5) described relation between the position of projector retina plant and unwrapped phase value, for any pixel of camera, its homologous point in the projector only relevant to its unwrapped phase value." L205 What degree of polynomial is used (i.e. what is the value of n in eq. 8)? L255 "anther": other? L259 It would be nice to have more tests on objects with known dimensions. L263 Table 1: values in mm? How do these values compare with similar test using different approaches using unwrapped phase? L298 "it only needs 1/6s to acquire": how do other systems fare in this respect? Is this time significantly shorter than that implementing other approaches? Do you anticipate any problem with occlusions, in particular because of the long(er) baseline between cameras and projector?
Author Response
Dear Reviewers:
Thank you for your letter and for the reviewer's comments concerning our manuscript “Title”(ID: sensors-587422 ).
Title: Fast 3D surface measurement with wrapped phase and pseudorandom image
Thoses comments are very helpful for us to revise and imporve our paper. Those comments are all valuable and very helpful for revising and improving our paper, as well as the importance guiding significance to our researches. We have studied these comments carefully and tried our best to revise and imporve the manuscript.
Revised portion are marked underline the paper.
English have been improved. Our main corrections in the manuscript and the responds to the reviewer's comments are as follows and a PDF file " Revised.pdf" ( main corrections are marked underline ).
Best wish
Xing Liu

Round 2
Reviewer 2 Report
With respect to the document named sensors-587422-coverletter.pdf
Answer 5
these points are usually called "homologous points", see your own definition at lines 88-89 of the revised paper: "pairs of image coordinates from the same object point (the homologous points)". Are they different?
Answer 6
Part of the text of the answer is missing, but the revised text contains the answer.
Answer 7
I know the pseudo image are generated by a software, my request was (is) of a very brief explanation of what pseudorandom image means.
Answer 14
I am sorry, but I do not understand your answer. "Yes it is.": *What* it is?
With respect to the revised paper (sensors-587422-peer-review-v2.pdf):
The language still needs checking, I wrote some suggestions but a comprehensive check is required.
L90 "Then triangulation geometry and the unwrapped phase are two significant conditions for 3D reconstruction based on unwrapped phase as described by Christian B[17], which obtain from the calibration system parameters and the fringe images recoded by cameras, respectively": "...which *are obtained* from..."
L93 "homologous" -> "homologous points"
L94 "only need to been" -> "only need to be"
L99 "The basic idea of the time phase unwrapping was to vary" -> "The basic idea of the time phase unwrapping is to vary"
L100 "Phase unwrapping was is then carried" -> "Phase unwrapping is then carried"
L104 "desire of a saving of the" -> "desire of saving the"
L126 "is make up a binocular stereovision" -> "create a binocular stereovision configuration"
L134 acronyms should be explained at their first use: FPP
L134 "ofactive" -> "of active"
L150 "C1with" -> "C1 with"
L150 "Then each pixel of the camera C1 with the wrapped phase, using the triangulation made up by the camera C1 and projector P to obtain candidate 3D points; by the aided of correlation of the camera C1 and C2, eliminate the false one and build the 3D point cloud.": is part of the sentence missing?
L171 "its homologous points on projector chip is located region" -> "its homologous points on projector chip are located region"
L172 "Fig3. . In this region" -> "Fig3. In this region"
L186 "The different possible corresponding point pairs P c1 ï‚« P c 2,i exist between two cameras." -> "Different possible corresponding point pairs P c1 ï‚« P c 2,i exist between two cameras."
L188 "the false is eliminated from possible corresponding point pairs." -> "false homologous are eliminated from possible corresponding point pairs."
L203 "which corresponding to real 3D point" -> "which corresponds to the real 3D point" (I would say it corresponds to the homologous point).
